# IgG and IgE Autoantibodies to IgE Receptors in Chronic Spontaneous Urticaria and Their Role in the Response to Omalizumab

**DOI:** 10.3390/jcm12010378

**Published:** 2023-01-03

**Authors:** Carlo Alberto Maronese, Silvia Mariel Ferrucci, Chiara Moltrasio, Maurizio Lorini, Vincenzo Carbonelli, Riccardo Asero, Angelo Valerio Marzano, Massimo Cugno

**Affiliations:** 1Dermatology Unit, Fondazione IRCCS Ca’ Granda Ospedale Maggiore Policlinico, 20122 Milan, Italy; 2Department of Pathophysiology and Transplantation, Università degli Studi di Milano, 20122 Milan, Italy; 3Department of Medical Surgical and Health Sciences, University of Trieste, 34137 Trieste, Italy; 4Dipartimento di Scienze Cliniche e di Comunità, Università degli Studi di Milano, 20122 Milan, Italy; 5Ambulatorio di Allergologia, Clinica San Carlo, Paderno Dugnano, 20037 Milan, Italy; 6Internal Medicine, Fondazione IRCCS Ca’ Granda Ospedale Maggiore Policlinico, 20122 Milan, Italy

**Keywords:** chronic spontaneous urticaria, omalizumab, autoimmunity, autoallergy, FcεRI

## Abstract

**Background:** Chronic spontaneous urticaria (CSU) is defined as the recurrence of unprovoked transient wheals and itch for more than 6 weeks. Currently, there is an unmet need concerning response prediction in CSU. The present study investigated biomarkers of type I and type IIb autoimmunity as potential predictors of response to omalizumab in CSU. **Materials and methods:** Differences in levels of IgG and IgE autoantibodies targeting the high- and low-affinity IgE receptors (FcεRI and FcεRII, respectively), as well as spontaneous and specifically triggered leukotriene C (LTC)4 release by basophils from the investigated subjects, were evaluated in 18 consecutive, prospectively enrolled CSU patients and 18 age- and sex-matched, healthy non-atopic controls. **Results:** The patients with CSU had higher levels of anti-FcεRI IgE (542 (386.25–776.5) vs. 375 (355–418), optical density (OD), *p* = 0.008), and IgG (297 (214.5–431.25) vs. 193.5 (118–275) OD, *p* = 0.004) autoantibodies relative to the controls. Simultaneous anti-FcεRI IgG and IgE positivity (i.e., both autoantibody levels above the respective cut-offs) was recorded only in late- and non-responders (3/8 and 1/2, respectively). **Discussion:** Significantly higher anti-FcεRI IgE autoantibody levels were found in the CSU patients as compared to the controls, supporting FcεRI as an autoallergic target of IgE (autoallergen) in the complex pathophysiological scenario of CSU. The co-occurrence of anti-FcεRI IgG and IgE autoantibodies was documented only in late- and non-responders, but not in early ones, crediting the co-existence of autoimmune and autoallergic mechanisms as a driver of late/poor response to omalizumab.

## 1. Introduction

Chronic spontaneous urticaria (CSU) is defined as the recurrence of unprovoked transient wheals and itch for more than 6 weeks [1]. CSU treatment has been reshaped by the advent of the anti-immunoglobulin (Ig)E monoclonal antibody, omalizumab. However, an unmet need still exists concerning response prediction [2]. Several biomarkers have been proposed as possible predictors of both response and non-response to second-generation antihistamine drugs, omalizumab and cyclosporine, with low total IgE being the single best predictor of non-response to omalizumab [3].

In addition to IgG autoantibodies directed against the high-affinity IgE receptor (FcεRI), whose involvement in CSU pathogenesis is well-recognized, IgE directed against a variety of autoallergic targets (autoallergens) [4,5] have been identified, including IgE anti-FcεRI [6].

The present study investigated biomarkers of type I and type IIb autoimmunity [7] as potential predictors of response to omalizumab in CSU. To this end, the differences in the levels of IgG and IgE autoantibodies targeting the high-and low-affinity IgE receptors (FcεRI and FcεRII, respectively), as well as spontaneous and specifically triggered leukotriene C (LTC)4 release by the basophils from the investigated subjects, were evaluated in 18 consecutive, prospectively enrolled CSU patients and 18 age- and sex-matched, healthy non-atopic controls.

## 2. Materials and Methods

Patients with severe CSU, i.e., Urticaria Activity Score (UAS)7 > 25, underwent subcutaneous omalizumab treatment at a dose of 300 mg every 4 weeks, following the indications of the European Academy of Allergy and Clinical Immunology (EAACI) guidelines [1]. The patients were evaluated at baseline and then every 4 weeks for 24 weeks after the omalizumab initiation, and UAS7 was recorded at each time point. An early response was defined as the achievement of a UAS7 score < 6 within 4 weeks after the first administration, and a late response was the achievement of the same reduction within 12 weeks after initiation, while non-response was the symptoms remaining unchanged 4 weeks after the third administration. The study was conducted according to the ethical principles of the Declaration of Helsinki and the code of Good Clinical Practice. The patients and controls gave informed written consent to the use of their sera and relative data. The local review board approved the study. 

Anti-FcεRI, anti-FcεRII IgE, and IgG autoantibodies were measured by a sandwich enzyme-linked immunosorbent assay at baseline, as described in detail elsewhere [6]. For cut-off levels, the highest value recorded among the healthy controls was adopted for each assay. A cellular antigen stimulation test (CAST) was used to evaluate the LTC4 release from the subjects’ basophils, both spontaneously and after stimulation with anti-IgE 10 mg/mL (Sigma, St. Louis, MO, USA), anti-FceRI 10 mg/mL (Bühlmann, Allschwil, Switzerland), or anti-FcεRII 10 mg/mL (Bühlmann, Allschwil, Switzerland), as previously described [8].

The results are expressed as the medians and interquartile ranges (IQR). The differences in terms of autoantibody levels and LTC4 release were evaluated by means of the Mann–Whitney U test between the CSU patients and the controls, and among the patients according to their response to the omalizumab. The correlations between the autoantibody levels and the LTC4 release were assessed by means of Spearman’s *rho*. *p*-values below 0.05, two-sided, were considered statistically significant (IBM SPSS Statistics for Windows, version 27.0, IBM Corp., Armonk, NY, USA).

## 3. Results

The demographics, clinical, and laboratory features of both groups are summarized in Table 1.

The patients with CSU had higher levels of anti-FcεRI IgE (542 (386.25–776.5) vs. 375 (355–418), optical density (OD), *p* = 0.008), and IgG (297 (214.5–431.25) vs. 193.5 (118–275) OD, *p* = 0.004) autoantibodies relative to the controls (Figure 1a,b).

Interestingly, the basophils from the CSU patients released significantly more LTC4 upon stimulation with anti-FcεRII antibodies than the basophils of the controls (2130.15 (535.325–3383.425) vs. 543.6 (304.5–854.5) pg/mL, *p* = 0.029) (Figure 1c). The basophils from the CSU patients also released more LTC4 upon stimulation with anti-FcεRI antibodies (746.35 (245.025–1072.075) vs. 357.85 (223.4–793.70 pg/mL, *p* = 0.183), albeit without reaching statistical significance.

Among the CSU patients, a statistically significant inverse correlation was documented between anti-FcεRII IgE autoantibody levels and LTC4 released from the patients’ basophils after in vitro stimulation with anti-FcεRII antibodies (r = −0.589, *p* = 0.010) (Figure 1d). Moreover, a similar trend was seen between anti-FcεRI IgE autoantibody levels and spontaneous LTC4 released from the patients’ basophils (r = −0.452, *p* = 0.06).

Simultaneous anti-FcεRI IgG and IgE positivity (i.e., both autoantibody levels above the respective cut-offs) was documented in the late- and non-responders (3/8 and 1/2, respectively), but not in the early responders to omalizumab.

## 4. Discussion

Herein, we found significantly higher anti-FcεRI IgE autoantibody levels in the CSU patients as compared to the controls. This seems to confirm FcεRI as an autoallergen in the complex pathophysiological scenario of CSU [9], with IgE autoantibodies directed against this target synergizing with the well-known IgG counterpart. Indeed, the co-occurrence of anti-FcεRI IgG and IgE autoantibodies in late- and non-responders, but not in early ones, credits the co-existence of autoimmune (i.e., type IIb autoimmunity) and autoallergic (i.e., type I autoimmunity) mechanisms as a driver of late/poor response to omalizumab.

Our data on the inverse correlations between LTC4 release in vitro from patients’ basophils and autoantibody levels may support the concept that autoantibodies may desensitize basophils to further activation. Future studies on the release of histamine, calcium, and β-hexosaminidase from basophils may be of help to confirm this view. In line with what we have observed, MacGlashan et al. demonstrated that the presence of autoantibodies to IgE or FcεRI is highly correlated with the non-responder phenotype of the basophil in CSU [10]. Although the present study differs from that by MacGlashan et al. in terms of experimental design [10], it could be hypothesized that the inverse correlation seen in our patients may reflect changes in the function of proteins involved in the signaling of IgE receptors, including spleen tyrosine kinase (SYK).

## 5. Conclusions

In conclusion, our findings, while confirming a *milieu* of co-existing type I and type IIb autoimmunity in the pathophysiology of CSU, underscore the challenges in finding reliable response predictors to omalizumab.

## Figures and Tables

**Figure 1 jcm-12-00378-f001:**
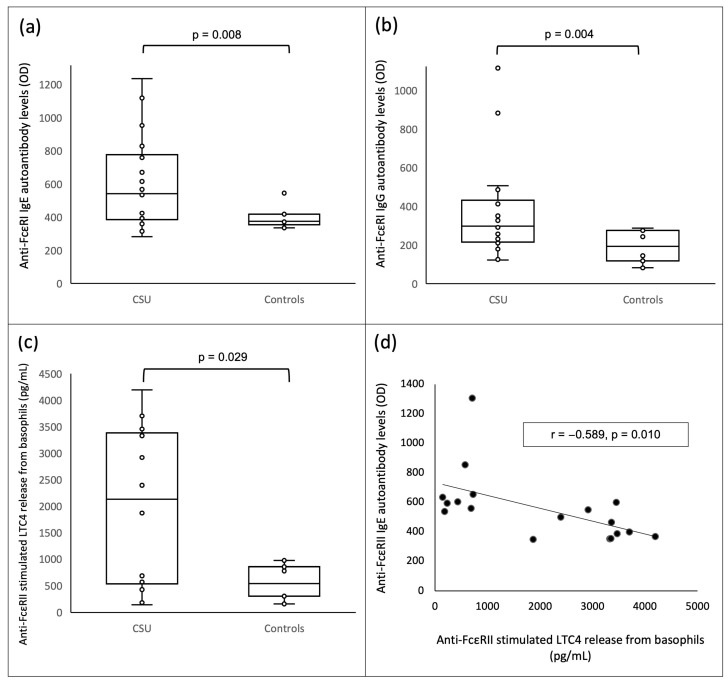
Box plots of anti-FcεRI IgE autoantibody levels: (**a**) anti-FcεRI IgG autoantibody levels, and (**b**) anti-FcεRII stimulated LTC4 release from basophils, (**c**) in patients with chronic spontaneous urticaria and in healthy non-atopic controls. (**d**) scatter plot showing inverse correlation between anti-FcεRII IgE autoantibody levels and anti-FcεRII stimulated LTC4 release. CSU = chronic spontaneous urticaria; OD = optical density.

**Table 1 jcm-12-00378-t001:** Clinical and laboratory features of chronic spontaneous urticaria (CSU) patients and healthy non-atopic controls. IgE = Immunoglobulin E; IQR = interquartile range; LTC4 = leukotriene C4; OD = optical density. Statistically significant *p*-values are in bold.

	CSU Patients (*n* = 18)	Healthy Non-Atopic Controls (*n* = 18)	*p*-Value
*Gender*			
Male, *n* (%)	6 (33.3%)	8 (44.4%)	-
Female, *n* (%)	12 (66.7%)	10 (55.6%)	-
Age (years), median (IQR)	52 (32.75–60.25)	49 (34–58)	-
Time from onset to omalizumab (months), median (IQR)	14.5 (8.25–63)	-	
Angioedema, *n* (%)	8 (44.4%)	-	-
*Response to omalizumab*			
Early response, *n* (%)	8 (44.4%)	-	-
Late-response, *n* (%)	8 (44.4%)	-	-
Non-response, *n* (%)	2 (11.2%)	-	-
anti-FcεRI IgE autoantibody levels (OD), median (IQR)	542 (386.25–776.5)	375 (355–418)	**0.008**
anti-FcεRII IgE autoantibody levels (OD), median (IQR)	544 (383–610.75)	505.5 (431–676)	0.775
anti-FcεRI IgG autoantibody levels (OD), median (IQR)	297 (214.5–431.25)	193.5 (118–275)	**0.004**
anti-FcεRII IgG autoantibody levels (OD), median (IQR)	314.5 (191–506.5)	346 (257–463)	0.775
Spontaneous LTC4 release (pg/mL), median (IQR)	70.5 (38.2–129.4)	89.1 (71.2–110.3)	0.296
anti-IgE stimulated LTC4 release (pg/mL), median (IQR)	724.1 (418.9–1104.5)	436.1 (295.6–1287.2)	0.587
anti-FcεRI stimulated LTC4 release (pg/mL), median (IQR)	746.35 (245.025–1072.075)	357.85 (223.4–793.7)	0.183
anti-FcεRII stimulated LTC4 release (pg/mL), median (IQR)	2130.15 (535.325–3383.425)	543.6 (304.5–854.5)	**0.029**

## Data Availability

All available information is contained within the manuscript.

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
