# Peer review of "IgG and IgE Autoantibodies to IgE Receptors in Chronic Spontaneous Urticaria and Their Role in the Response to Omalizumab"

_jcm, 2023, doi:10.3390/jcm12010378_

Round 1
Reviewer 1 Report
I think this article is interesting and well-written.
Author Response
We would like to thank the Reviewer for the positive comments and the appreciation of our work.
Reviewer 2 Report
This study has investigated the role of IgG and IgE autoantibodies to IgE receptor 1 that may or may not interfere during omalizumab (a monoclonal antibody to soluble IgE ) used for the treatment of spontaneous urticaria or hives. The authors have shown the presence of IgG and IgE auto antibodies and the release of leukotriene C4 in patients with urticaria. They have also shown how the presence of these autoantibodies may interfere with Omalizumab treatment. Similar studies have been reported earlier and the authors' report is a confirmation of the problems in the treatment of hives we face. This is an important addition to the field and reiterates the urgent need of developing better diagnostics and therapeutics for this common but severe allergic condition. This manuscript may be accepted for publication.
The authors looked at the release of LTC4 for functional assay of basophils. Did they look at other factors like histamine levels, calcium release and β-hexosaminidase from basophils.
Author Response
We thank the Reviewer for the positive comments and the appreciation of our work. We agree with the Reviewer that investigating the release of histamine, calcium and β-hexosaminidase from basophils may be of help to support our view. For this reason, we added a comment in the discussion on page 4 lines 120-122.
Reviewer 3 Report
Two minor points. It is not clear that "autoallergen" is an appropriate term for Fcepsilon RI.... allergen is normally considered a foreign protein.
It is an auto-immune response. Understanding the difference and value of omalizumab treatment in some patients is useful. And the amount of and specificity of antibodies is the markers is important.
The authors cited MacGlashan 2021, but did not connect the findings of MacGlashan to what might be going on with the patients in their Italian study. While they obviously could not have tested Syk in this study without including it initially, they could mention it as a possibilitiy.
Author Response
We thank the Reviewer for the positive comments and the appreciation of our work.
1) The concept of autoallergy, also known as type I autoimmunity, forms an integral part in the current landscape of CSU pathogenesis. Please, see: Kolkhir P, Muñoz M, Asero R, et al. Autoimmune chronic spontaneous urticaria. J Allergy Clin Immunol. 2022;149(6):1819-1831. doi: 10.1016/j.jaci.2022.04.010.
As reported in the above-mentioned article, self-antigens targeted by specific IgE auto-antibodies may also be called autoallergens.
However, we agree with the Reviewer that allergens are normally foreign proteins, so we decided to better explain the concept in our manuscript by using “autoallergic target (autoallergen)” (on line 27, in the abstract, and on line 44, in the introduction).
2) The study by MacGlashan et al. showed the existence of an inverse correlation between the presence of autoantibodies and basophil responses, meaning that autoantibodies lead to a progressive desensitization of basophils. Although we did not perform in depth investigations into Syk function, we felt that the inverse correlation we found between autoantibody levels and LTC4 release might be in line with MacGlashan’s findings. It must be emphasized that our study evaluated autoantibody levels and LTC4 release at baseline and categorized these results based on clinical response. Differently, MacGlashan et al. studied basophil dynamics before and after treatment with omalizumab in CSU patients.
As suggested, the following sentence was added on page 5 lines 124-127 to comment on the possibility of Syk involvement in our patients too: “Although the present study differs from that by MacGlashan et al. in terms of experimental design, it could be hypothesized that the inverse correlation seen in our patients may reflect changes in the function of proteins involved in the signalling of IgE receptors, including spleen tyrosine kinase (SYK).”.